# The Preventive Child and Youth Healthcare Service in the Netherlands: The State of the Art and Challenges Ahead

**DOI:** 10.3390/ijerph19148736

**Published:** 2022-07-18

**Authors:** Yvonne T. M. Vanneste, Caren I. Lanting, Symone B. Detmar

**Affiliations:** 1Dutch Knowledge Centre for Youth Health, NCJ, Churchilllaan 11, 3527 GV Utrecht, The Netherlands; 2Netherlands Organisation for Applied Scientific Research TNO, Department of Child Health, TNO, Schipholweg 77, 2316 ZL Leiden, The Netherlands; caren.lanting@tno.nl (C.I.L.); symone.detmar@tno.nl (S.B.D.)

**Keywords:** child and youth healthcare, preventive care, child health

## Abstract

The Netherlands has a unique system for promoting child and youth health, known as the preventive Child and Youth Healthcare service (CYH). The CYH makes an important contribution to the development and health of children and young people by offering (anticipatory) information, immunisation, and screening, identifying care needs and providing preventive support to children and their families from birth up to the age of 18 years. The CYH is offered free of charge and offers basic preventive care to all children and special preventive care to children who grow up in disadvantaged situations, such as children growing up in poverty or in a family where one of the members has a chronic health condition. Basic care is supported by 35 evidence-based guidelines and validated screening tools. Special care is supported by effective interventions. The impact of the CYH is high. It is estimated that every EUR 1 spent on the CYH provides EUR 11 back. Although the Dutch CYH is a solid public health system with a reach of up to 95% among young children, the access to this service could be further improved by paying more attention to health literacy, making special care available to all children in need and improving transmural and integrated care coordination. In addition, the generation of nationwide data could help to demonstrate the impact of the CYH and will direct and prioritise the necessary care. By continuously developing care on the basis of new (scientific) insights and (societal) issues, the CYH will continue to offer all children in the Netherlands the best preventive healthcare.

## 1. Introduction

The Convention on the Rights of the Child states that “Every child has the right to the highest attainable standard of health and to a safe place to live and play” [1]. These rights provide the basis and direction for the preventive Child and Youth Healthcare service (CYH) in the Netherlands. The CYH makes an important contribution to the development and health of children and young adults by offering (anticipatory) information, immunisation, and screening, identifying care needs and providing support to children and their families from birth up to the age of 18 years. Every child who lives in the Netherlands is entitled to care from the CYH, and it is the responsibility of the CYH to provide expert advice in the best interests of the child’s health. The purpose of this study is to describe how the CYH is organised in the Netherlands and what challenges it faces. It includes a state-of-the-art review based on the study of the literature. The discussion section considers a number of challenges currently faced by the CYH, both with regard to the implementation of services as well as care delivery, and in regard to a number of new problems that may arise and which need to be dealt with by policymakers and professionals within the CYH.

## 2. Three Basic Principles

To be able to provide expert advice and the best care, the CYH is guided by three basic principles: the biopsychosocial-ecological perspective on health and development, a joint assessment, and shared decision-making by professionals, children and their parents. Firstly, the biopsychosocial-ecological model reflects the complex interaction of biological–genetic, psychological, social and environmental factors related to a child’s development and health [2]. The interaction between these factors plays a role in the origin and in the course of many developmental and health problems. Secondly, in order to adequately assess care needs, it is crucial that children and their parents are actively engaged in the assessment and decision-making process through, for example, using the GIZ methodology [3]. The GIZ is based on the Common Assessment Framework triangle (CAF) (Figure 1), which creates a clear visual substantive framework for a joint assessment consultation [4]. Thirdly and lastly, when faced with the task of making decisions about the care to be delivered, professionals share the best available evidence with children and their parents and support them in considering options and achieving informed preferences (shared decision-making) [5].

In order to share the best available options with children and their parents, i.e., options that are as scientifically substantiated as possible, most appropriate according to the professional, and feasible and acceptable for the child and his/her parents, professionals are supported by 35 evidence-based guidelines and several validated screening tools. The professional guidelines are owned by the professional associations and published by the NCJ, the Dutch Knowledge Centre for Youth Health (Nederlands Centrum Jeugdgezondheid).

## 3. Economic, Population, Societal and Health Characteristics

The Netherlands is a densely populated (in 2020, 517 people per km^2^) country with just over 17 million inhabitants, of whom 21.7% are below the age of 20 (Table 1) [6]. It is ranked the fourth-richest country in the European Union. In 2020, the gross domestic product (GDP) per capita amounted to EUR 45,793. The Netherlands spends about 10% of its GDP on healthcare, which is the tenth highest expenditure in Europe, and just under 2% of the GDP is spent on prevention [7]. The Netherlands has one of the best digital infrastructures in the world. In 2020, 97% of households in the Netherlands had Internet access [6].

Despite the fact that the Netherlands is among the most prosperous and egalitarian countries in the EU, the gap between the ‘haves’ and the ‘have-nots’ is widening. That is, in the last two decades, the part of the Dutch adult population that is considered at risk of poverty and/or social exclusion has increased from 10% to 16% [8]. Moreover, in 2019, a striking 57% of Dutch adults experienced chronic disease [9], 18% have a mental health disorder [10], and an increasing percentage of the Dutch adult population is overweight (35% in 1990, 51% in 2020) [11]. Although the number of smokers is declining, 20% of Dutch adults still smoke on a daily basis [11], and an estimated 9% of Dutch women continue to smoke while pregnant [12]. Increasing numbers of children are born into single-parent families: the number rose from 6% in 2000 to 9% in 2020 [13].

## 4. Child and Youth Health

In 2020, over 164 thousand children were born in the Netherlands. Life expectancy at birth is steadily increasing. Meanwhile, one in six children (15%) are born prematurely (<37 weeks of gestation) and/or small for gestational age (Table 1) and are thus at risk of serious short- and long-term health problems [13]. Moreover, one in six Dutch children and young adults (15%) are overweight or obese. This is in line with the rising rates of overweight and obesity worldwide [14]. Nowadays, 26% of children and young adults (0–25 years) living in the Netherlands suffer from a chronic disease [15]. One of these chronic diseases is depression. Depression, together with anxiety, makes up about one-third (28.3%) of all causes of chronic disease among children and young adults [16]. Government measures introduced to combat the COVID-19 pandemic have presumably worsened the situation regarding mental health issues [17].

In 2020, van Berkel et al. computed the prevalence of child maltreatment. This was found to be at a rate of 26–37 per 1000 children (i.e., 89,160 to 127,190 children and young adults each year, or 3.2% of all minors), with educational/emotional neglect being the most prevalent, followed by physical neglect [18]. Research based on the self-reports of children presents us with even more striking figures. For example, one-quarter (27%) of all 11 year olds in Dutch elementary schools reported maltreatment by someone from either within or outside their families at least once in their lives [19].

Despite these numbers, Dutch children and young adults are among the tallest, the happiest, and the healthiest in the world [15,20]. When asked, young people in the Netherlands report high levels of satisfaction in terms of social embeddedness. In a WHO survey, more than 75% of Dutch children and young adults said that they felt (strongly) supported by their parents and could relate to them in an easy manner [15]. In the same survey, Dutch youths rated their lives 7.3 at age 15, and 8.4 at the age of 11, on a scale of 0 to 10 (with 0 meaning ‘the worst life that I can imagine’, and 10 ‘the best life I can imagine’). They also reported considerably less psychosomatic disorders (such as headaches and sleeping problems) than their European counterparts. Lastly, this survey found the Netherlands to have the lowest percentage of problematic social media users. A Dutch study showed that over 67% of Dutch children and young adults felt (strongly) supported by their friends and classmates [21].

## 5. Preventive Child and Youth Healthcare

The CYH sector is organised nationally and implemented locally through 38 regional CYH organisations or public health subdivisions that come under the responsibility of, and are funded by, the municipalities. The CYH is offered free of charge and has a reach of up to 95% among young children [22]. It offers basic preventive care to all children aged 0–18 and special preventive care to children who grow up in disadvantaged situations. The municipalities provide basic care and receive funding for this via the central government. Interventions intended as special care for vulnerable groups are not structurally financed by the central government. Municipalities are expected to free up funding from their own resources. The level of funding as well as the choice of interventions to be implemented differ as per the municipality.

The CYH is organised and funded differently from the curative healthcare sector. CYH professionals are specialised in prevention and, as such, do not treat children or hand out prescriptions. Unlike referrals for curative somatic care, which CYH professionals can only provide through the general practitioner (GP) of the child, they can refer directly to child welfare and mental healthcare services. Like the CYH, child welfare and mental healthcare services are financed by the municipalities, as opposed to the curative sector, which is funded via private insurance companies. Most municipalities have made regional agreements about the referral process from the CYH to child welfare and mental healthcare services.

CYH professionals work in multidisciplinary teams with specialised physicians, nurses and physician assistants, and occasionally with dietitians, speech therapists or educationalists. In addition, CYH professionals work closely together with care providers, such as midwives and GPs, and professionals in nurseries, childcare and schools to identify and support children in need. If a child is not reached by the CYH for basic care, a non-response protocol comes into effect, including more outreach activities. In addition, there is a national guideline for reaching and guiding refugee children. The GP is informed about children who are ultimately not reached by the CYH.

### 5.1. Basic Child and Youth Healthcare

Basic care provided by the CYH is described in the Public Health Act of 2008 [23], which was set up to protect and promote the development and physical and mental health of all children, at both the individual and the population level. At the individual level, basic care is achieved by offering all children and their parents individual, tailor-made consultations which are initiated by the CYH organisation. The basis is a minimum number of 15 contacts in order to enable the provision of the care to which the child is entitled. If necessary, and in consultation with the children and their parents, more care can be provided, including short-term support and guidance. It is up to the CYH to ensure that the children and their parents receive the necessary care. The Public Health Inspectorate monitors the quality of the care provided. Basic care is provided through face-to-face consultation in health centres, schools or at home, and by telephone or online contact. In addition, parents and young people can initiate contact with the CYH via chat, telephone or WhatsApp, or by emailing their CYH organisation.

Basic care includes strengthening and preserving the health and development of children, thereby preventing problems later on in life. It also includes the implementation and monitoring of the National Vaccination Programme and national screening programmes, such as the Neonatal Screening Programme, screening for vision disorders and congenital hip dysplasia, the prevention of sudden infant death syndrome (SIDS), and the provision of advice on the prevention of accidents in and around the home. The regular consultations also include advice on a healthy lifestyle and parenting issues. Extra attention is paid to children with an increased risk of health problems, such as children growing up in poverty or in a family where one of the members has a chronic health condition. Growing up in a disadvantaged situation can cause toxic stress, also known as early life stress (ELS). ELS can partly explain the impact of these situations on the health of children, and can also increase the risk of child abuse, long-term absenteeism and school dropout. During the European Union for School and University Health and Medicine (EUSUHM) congress in 2019, the Dutch CYH joined in the European Union’s commitment to reducing the impact of ELS by the early identification of ELS, promoting the mentalizing capacity of parents and other adults around the child, and fostering children’s resilience [24].

At the population level, as a result of modern digital developments, reliable preventive health information that is appropriate for the age of a child or young person is offered on a 24/7 basis via websites and applications. Online platforms can help children and young people to find the most appropriate care for their specific needs further supported by e-health tools, such as devices that support adequate health behaviour and improve lifestyle. In addition, policy advice is provided based on the collective data, including biometric as well as psychological and environmental data, provided by CYH professionals who keep individual (electronic) records of the care provided. On the basis of these data, CYH professionals, in collaboration with epidemiologists, provide policy advice to municipalities and schools. The CYH promotes and supports the implementation of evidence-based interventions and health-promoting programmes aimed at specific groups of children, such as those at school. Most programmes are age-appropriate and intended for all children, as opposed to interventions intended as special care for vulnerable groups. Examples include programmes that focus on a healthy lifestyle, sex education programmes and resilience-building programmes. These programmes are also part of the Healthy School Programme, which helps schools to work purposefully and efficiently on the health of all their pupils [25].

### 5.2. Special Care

Offering substantial further support through the deployment of non-regular interventions is not part of basic care. As mentioned earlier, interventions intended as special care for vulnerable groups are not structurally financed by the central government. A national database run by the National Institute for Public Health and the Environment (RIVM) of the Ministry of Health, Welfare and Sport provides 24 evidence-based interventions that can be used by professionals in the CYH [26]. Some of the more frequently used interventions are managed by the NCJ, which is committed to the quality of their implementation and further development. Examples of such interventions include small-group parenting courses and parenting support through video-feedback interventions that aim to promote positive parenting and sensitive discipline [27], and interventions such as the Nurse Family Partnership, in the Netherlands known as ‘VoorZorg’, for very vulnerable and often young first-time mothers. VoorZorg is an effective evidence-based intervention for reducing prenatal stress as well as lifestyle and parenting problems and the prevention of child abuse, which is offered on indication by midwives [28]. In order to prevent young people from dropping out of school, the intervention ‘Medical Advice for Sick-reported Students’, abbreviated as MASS, is offered to all children in vocational and secondary education. MASS aims to optimise students’ health and maximise students’ participation in school activities and has been shown to be effective in reducing sickness absence rates in secondary schools [29] and vocational education [30]. MASS is currently being developed and evaluated for primary education [31].

## 6. Impact and Returns

Several studies have demonstrated the long-term impacts of adverse childhood experiences faced by children growing up in (educationally) disadvantaged situations on a range of health outcomes in adulthood, including mental health, problematic alcohol use, heart disease [32,33], educational outcomes and earnings [34] and life expectancy [35]. According to several international studies, early intervention during childhood can be an effective strategy for preventing problems later on in life, as well as intergenerational inequality [36,37,38]. The earlier the intervention is implemented, the higher the expected returns are [37]. The costs of the Dutch preventive CYH have been estimated at EUR 155 million per year [39], whereby every EUR 1 invested in the CYH is estimated to yield EUR 11 in return [40]. Although it can be assumed that the impact of the Dutch CYH is considerable, given its high reach (more than 95%) and the application of several interventions that have been proven effective in the Netherlands, the evidence showing the overall impact of the Dutch CYH on the health and development of children and young people is scarce. One recent study showed that the previously established CYH SIDS guideline, promoting the supine sleeping position for babies and the use of their own beds or cribs, decreased infant mortality significantly [41,42].

## 7. Discussion/Reflection

Despite the relative success of the preventive CYH service, as described in this paper, a number of challenges for the CYH need to be considered. Such challenges consist of equal access to preventive CYH services, as well as its care delivery, the need for the coordination of transmural care and the use of CYH data in a national database. Moreover, there are new problems, such as migration and climate change, that may arise and that need to be dealt with by the CYH.

### 7.1. Access for All

Although the reach of the CYH is very high, children who grow up in disadvantaged families are typically less well reached than others [22]. About one in four people, especially those of low socio-economic status and education, have been shown to have limited health literacy (insufficient, 1.8%; problematic, 26.9%) [43]. We know from previous studies that low health literacy has a major impact on access to, as well as the use and effectiveness of, health care, and is associated with (early) morbidity and mortality and additional health care costs [44]. All of this explains why low health literacy is a major contributor to rising inequalities in health, well-being and participation. For the CYH, promoting health literacy is an important strategy for increasing its reach and effectiveness, which is why this issue is receiving increasing attention from policymakers and CYH professionals. Strangely enough, the previously mentioned special care interventions aimed at at-risk groups are not offered by all municipalities in the Netherlands. As a result, not all children and their parents who need this extra support have access to it. To increase the effectiveness of the CYH, it is necessary to make evidence-based interventions equally accessible to all groups of children. The effectiveness of the preventive CYH service could be increased by integrating the CYH earlier into (future) parents’ lives, i.e., during pregnancy. From June 2022, it will be mandatory for CYH professionals to offer prenatal home visits to pregnant women in vulnerable situations. However, the CYH is convinced that all pregnant women need to have access to this option.

### 7.2. Coordinating Care

Due to the increase in the number of children with chronic diseases and complex long-term conditions, the coordination of transmural and integrated care is becoming increasingly important. These children often require not only medical care, but also support outside the hospital environment with respect to prevention, care and well-being, because of the impact that a chronic condition can have on the well-being and functioning of the child, the family and the child’s functioning at school. It is important that children and their parents are supported in formulating their care needs and in arranging and coordinating all necessary care and guidance. CYH professionals can take on this role, as coordinating health professionals like no other, because of their biomedical and psychosocial expertise, their focus on the context of the child, their contacts with childcare and school professionals and their intensive collaboration and presence in the neighbourhood.

### 7.3. Insights from the Use of CYH Data in a National Database

CYH data are currently only available at a regional level. During the COVID-19 pandemic, it became once again clear how important it is that CYH data can be generated on a national level. By linking these regional data to a national database, more insight into the health and well-being of children and young adults can be obtained in order to prioritise children in the greatest need of care, to demonstrate the effects of the CYH and to support the implementation of effective child and youth healthcare policy, not only during a pandemic but also in the long-term. Indicators have recently been drawn up that form the basis for a joint and unambiguous registration, as a prelude to the national upscaling of CYH data [45].

### 7.4. Implications for the Future

Future societal developments, such as those resulting from ever-increasing digitalisation, migration, changes in family composition and climate change, will certainly have consequences for the health and well-being of children. The CYH will have to prepare for this, and, here too, the biopsychosocial-ecological perspective is helpful for setting out actions. The experience and insights gained from the recent COVID-19 pandemic, which show the importance of fighting and preventing infectious disease, could lead to more attention to, and funding for, preventive activities.

## 8. Conclusions

The Dutch CYH is a solid public health system that offers basic and special preventive care and reaches more than 95% of children. Its impact could be even further improved by giving more attention to low health literacy, by making special care interventions accessible to all children in need and by taking on the role of coordinating health professionals in transmural care. The generation of nationwide data could help to show the impact of CYH and will direct and prioritise the necessary care. By continuously developing care on the basis of new (scientific) insights and (societal) issues, the CYH will continue to offer all children in the Netherlands the most optimal preventive healthcare.

## Figures and Tables

**Figure 1 ijerph-19-08736-f001:**
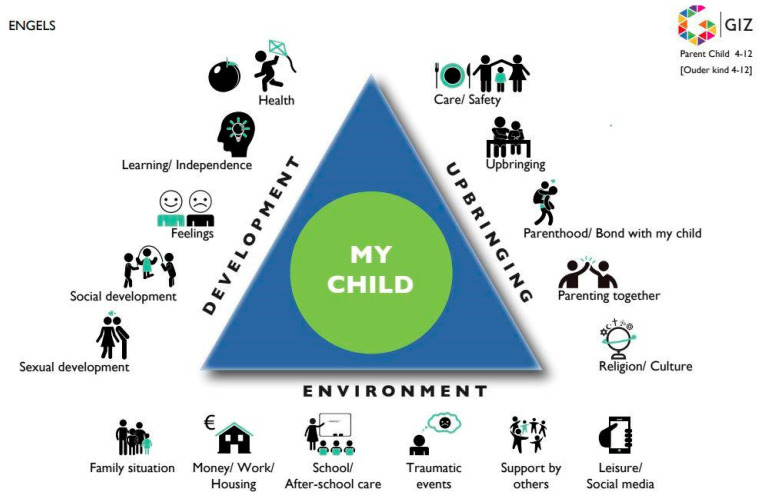
Common Assessment Framework (CAF). Available online: https://ars.els-cdn.com/content/image/1-s2.0-S0190740920321472-gr1a_lrg.jpg (accessed on 29 April 2022).

**Table 1 ijerph-19-08736-t001:** The Netherlands: population, health, and economic characteristics, 2000–2020 [6].

Indicator	2000	2010	2020
**Population**			
Total number (in millions)	15.9	16.6	17.4
Density (people per km^2^)	468	491	517
Live births (n)	206,619	184,397	164,262
Children and young adults in the ages of 0–20 y (% of total)	24.4	23.7	21.7
Life expectancy at birth (y)	78.1	80.8	81.4
Non-western migrants (% of total Dutch population)	8.9	11.2	13.7
People at risk of poverty and/or social exclusion (% of total Dutch population) ^a^	10.4	6.6	15.8
**Health**			
Overweight:			
% of adult population	44.3	48.2	51.0
% of children and young adults in the ages of 4 to 20 years	11.8 ^a^	13.6	15.1
Smokers (% of ages > 12 y)	32.0	26.3	20.1
Preterm births (<37 weeks) and/or SGA births (< p 10) (% of live births)	8	8	15
Immunisation rate:			
DTaP-IPV (% of children in the ages of 12–23 months)	95.6	95.0	93.1
MMR (% of children in the ages of 12–23 months)	95.6	96.2	93.6
**Economics**			
GDP (in 1000 euros per capita)	40	44	46
Health expenditure:			
% of GDP	7.7	10.2	11.2
in euros per capita	2187	3907	5137

^a^ People who live in a household with an income below the European poverty threshold (which is less than 60% of the national median disposable income); SGA small-for gestational age; DTaP-IPV, diphtheria, tetanus, pertussis and polio; MMR, mumps, measles, rubella; GDP, gross domestic product.

## Data Availability

There was no data collected for this study.

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
