# Peer review of "The Preventive Child and Youth Healthcare Service in the Netherlands: The State of the Art and Challenges Ahead"

_ijerph, 2022, doi:10.3390/ijerph19148736_

Round 1
Reviewer 1 Report
This descriptive paper presents an overview of the Preventive child and youth healthcare service in the Netherlands. The paper is well written but the following should be considered:
1. The purpose of the report is never clearly stated.
2. More details should be provided about the funding for the Preventive child and youth healthcare service. Do all municipalities contribute the same proportion of their budgets to the “basic care” of the service? Are municipalities freeing up funding for vulnerable groups or are their problems with this funding arrangement?
3. No information is provided about children from immigrant or racialized communities and this topic should be covered given the current concerns about inequities in healthcare.
Author Response
This descriptive paper presents an overview of the Preventive child and youth healthcare service in the Netherlands. The paper is well written but the following should be considered:
Point 1: The purpose of the report is never clearly stated.
Response 1: Many thanks for the comment. That is indeed missing and we have added the information about the purpose of the study after line 36: The purpose of this study is to describe how the CYH is organised in the Netherlands and to what challenges it faces.
Point 2: More details should be provided about the funding for the Preventive child and youth healthcare service. Do all municipalities contribute the same proportion of their budgets to the “basic care” of the service? Are municipalities freeing up funding for vulnerable groups or are there problems with this funding arrangement?
Response 2: We've added information about this after line 122: The level of funding as well as the choice of interventions to implement differ per municipality.
In the discussion section (Access for all, lines 238-242) we state that the funding arrangements are a challenge: Strangely enough, the previously mentioned special care interventions aimed at risk groups are not offered by all municipalities in the Netherlands. As a result, not all children and their parents who need this have access to this extra support. To increase the effectiveness of the CYH it is necessary to make evidence-based interventions equally accessible for all groups of children.
Point 3: No information is provided about children from immigrant or racialized communities and this topic should be covered given the current concerns about inequities in healthcare.
Response 3: Basic preventive care is offered to all children aged 0-18. The CYH reaches more than 95%. At regional level there is insight in the reach per group. However, at national level these data are lacking, as mentioned in the section Insights by use of CYH data in a national database: CYH data is currently only available at regional level. The regional CYH organisation knows which children are not being reached and acts on this by executing a non-response protocol including more outreaching activities. There is a national guideline on how to reach and guide refugee children.
We have added this information after line 136: If a child is not reached by the CYH for basic care, a non-response protocol comes into effect including more outreaching activities. In addition, there is a national guideline for reaching and guiding refugee children. The GP is informed about children who are ultimately not reached by the CYH.
.
In the section Access for all we state: Although the reach of CYH is very high, children who grow up in disadvantaged families are typically less well reached than others (22). As limited health literacy has a major impact on access to health care, across specific groups of people, CYH focuses on promoting health literacy.
Reviewer 2 Report
The focus of this paper is significant to the field of Child and Youth healthcare, however, the paper, in its current form, requires a lot of additional work.
The tithe did not reflect the nature of the paper, and the content did not allow access to the impact of the preventive Child and Youth Healthcare service (CYH). The paper only makes a reflection based on some indicators.
The main problem is that the reader cannot access the aim of the study and the content is presented without a visibly defined alignment. The author(s) needs the purpose of the study, the nature, and the methodology used to answer the study's purpose. It seems to be a reflection on preventive Child and Youth Healthcare service (CYH) but the purpose is not explained to the reader. I suggest clearly defining the purpose of the study and what methodology was used to answer the purpose (e,g, documental analysis,….). In terms of the sections, attending to the content presented, I suggest organizing the paper to answer three main questions: what does the legislation say? What does the data (statically data) say? What did the empirical studies say?
Author Response
The focus of this paper is significant to the field of Child and Youth healthcare, however, the paper, in its current form, requires a lot of additional work.
Comment: The title did not reflect the nature of the paper, and the content did not allow access to the impact of the preventive Child and Youth Healthcare service (CYH). The paper only makes a reflection based on some indicators. The main problem is that the reader cannot access the aim of the study and the content is presented without a visibly defined alignment. The author(s) needs the purpose of the study, the nature, and the methodology used to answer the study's purpose. It seems to be a reflection on preventive Child and Youth Healthcare service (CYH) but the purpose is not explained to the reader. I suggest clearly defining the purpose of the study and what methodology was used to answer the purpose (e,g, documental analysis,….). In terms of the sections, attending to the content presented, I suggest organizing the paper to answer three main questions: what does the legislation say? What does the data (statically data) say? What did the empirical studies say?
Response: Many thanks for the comment. The paper is a viewpoint and includes a state-of-the-art review through literature study.
First, we changed the title: Preventive Child and Youth Healthcare service in The Netherlands: State of the Art and Challenges Ahead.
Second, we added the purpose and the nature and an outline of the paper, after line 36: The purpose of this study is to describe how the CHY is organised in the Netherlands and to what challenges it faces. It includes a state-of-the-art review through literature study. The discussion section considers a number of challenges currently faced by the CYH, both with regard to the implementation of services as well as care delivery, and a number of new problems that may arise and which need to be dealt with by policymakers and professionals within the CYH.
Reviewer 3 Report
The system of preventive care for children and youth in the Netherlands is unique, based on many years of practice and tradition of the Netherlands. At the end of the second millennium, the WHO noted that three basic factors had substantially reduced child morbidity and mortality in the second half of the 20th century. These were vaccination, antibiotics and primary pediatric care. The latter is not in the Dutch system. In the countries of the European Union, there are three comprehensive health care systems in about a third of the following: a. / Pediatrician in primary care, b. / curative care at general practitioner and preventive care separate, and c. / mixed system a. / + b. /. Dutch is in category b. These systems are based on the country's tradition of care development and should incorporate the basic principles set out in the manuscript. However, it is clear from the presented results that not all measures achieve the desired effect. Please comment:
1./ The percentage of prematurity is high (above 8%), stratification of care for high-risk pregnancies is not mentioned.
2./ Obesity, chronic illness (26%) and depression are signals of the reserves of prevention programs.
Nevertheless, Dutch youth is the tallest, happiest and healthiest in the world. It is in a seeming contradiction to the above problems. The weakness is the fact that the program is managed regionally from the municipality and this can lead to differences in the quality of care.
3./ The manuscript does not state who is the main responsible coordinator of the whole system - it should be a pediatrician focused on community and social care. There is lacks information on how to solve the identified incorrect procedures and mistakes.
Author Response
The system of preventive care for children and youth in the Netherlands is unique, based on many years of practice and tradition of the Netherlands. At the end of the second millennium, the WHO noted that three basic factors had substantially reduced child morbidity and mortality in the second half of the 20th century. These were vaccination, antibiotics and primary pediatric care. The latter is not in the Dutch system. In the countries of the European Union, there are three comprehensive health care systems in about a third of the following: a. / Pediatrician in primary care, b. / curative care at general practitioner and preventive care separate, and c. / mixed system a. / + b. /. Dutch is in category b. These systems are based on the country's tradition of care development and should incorporate the basic principles set out in the manuscript. However, it is clear from the presented results that not all measures achieve the desired effect.
Many thanks for the comment. As the reviewer rightly points out, primary pediatric care is not included in the Dutch CYH (lines 123-127: The CYH is separately organised and funded from the curative healthcare sector. CYH professionals are specialised in prevention and as such do not treat children or hand out prescriptions. Unlike referrals to curative somatic care which CYH professionals can only provide through the general practitioner (GP) of the child, they can refer directly to child welfare and mental healthcare services.).
Comment 1. The percentage of prematurity is high (above 8%), stratification of care for high-risk pregnancies is not mentioned.
Response 1: As mentioned in the paper, CYH professionals work closely together with midwives and GPs to identify and support (future) children in need. See also lines 132-136: Therefore, CYH professionals work in multidisciplinary teams with specialised physicians, nurses and physician assistants, and occasionally dietitians, speech therapists or educationalists. In addition, CYH professionals work closely together with care providers such as midwives and GPs, professionals in nurseries and childcare and schools to identify and support children in need.
As stated in Access for all, the effectiveness of the preventive CYH could be increased by integrating the CYH earlier in (future) parents’ lives i.e. during pregnancy. From June 2022 it will be mandatory to offer prenatal home visits by CYH professionals to pregnant women in a vulnerable situation. However, CYH is convinced that all pregnant women need to have access to this option. (lines 242-246)
Comment 2: Obesity, chronic illness (26%) and depression are signals of the reserves of prevention programs. Nevertheless, Dutch youth is the tallest, happiest and healthiest in the world. It is in a seeming contradiction to the above problems. The weakness is the fact that the program is managed regionally from the municipality and this can lead to differences in the quality of care.
Response 2: Although CYH is implemented locally, the quality of care is guaranteed by 35 national evidence-based guidelines and validated screening tools. Special care is supported by effective interventions. The weakness of the system is that special care, although proven to be effective, is not offered to children in need by all municipalities. In the section ‘access for all’ we pay attention to this challenge for the Dutch CYH. (lines 238-242)
As mentioned in the discussion section, one of the challenges to address is also coordinating transmural and integrated care due to the increase in the number of children with chronic diseases and complex long-term conditions and the diversity of care that this also requires outside the hospital. (lines 247-258)
Comment 3: The manuscript does not state who is the main responsible coordinator of the whole system - it should be a pediatrician focused on community and social care. There is a lack of information on how to solve the identified incorrect procedures and mistakes.
Response 3: Basic care provided by the CYH is described in the Public Health Act of 2008 [23] which was set up to protect and promote the development and physical and mental health of all children, both at individual and population level. (See lines 138-140)
Article 22 of the Constitution stipulates that the central government, assigned to the Ministry of Health, Welfare and Sport, shall take measures to promote public health, resulting in Public healthcare. The CYH is part of Public healthcare. The central government and the municipalities are jointly responsible for the quality of care. In the Public Health Act it is written what basic care is offered to all children to protect and promote health and to prevent disease. The PHA also obliges the municipalities to maintain a CYH service for the provision of the CYH. The Ministry is advised on this by, among others, the Regional Public Health services and the National Institute for Public Health and the Environment (RIVM). Control of the quality and output of the CYH is outsourced by the government to the Public Health Inspectorate.
We’ve added information about this line 146: The Public Health Inspectorate monitors the quality of the care provided.